# Level of dietary adherence and determinants among type 2 diabetes population in Ethiopian: A systemic review with meta-analysis

**Teshager Weldegiorgis Abate**[1]*, **Minale Tareke**[2o], **Selam Abate**[3o], **Abebu Tegenaw**[1‡], **Minyichil Birhanu**[4], **Alemshet Yirga**[1‡], **Mulat Tirfie**[5‡], **Ashenafi Genanew**[6o], **Haileyesus Gedamu**[1‡], **Emiru Ayalew**[1‡]

**1** Department of adult health Nursing, School of Health Science, College of Medicine and Health Sciences, Bahir Dar University, Bahir Dar, Ethiopia, **2** Department of Psychiatry, School of Medicine, College of Medicine and Health Sciences, Bahir Dar University, Bahir Dar, Ethiopia, **3** Department of Health Officer, Merawi Primary Hospital, Amhara Health Bureau Dar, Ethiopia, **4** Department of Pediatric and Child Health Nursing, School of Health Sciences, College of Medicine and Health Sciences, Bahir Dar University, Bahir Dar, Ethiopia, **5** Department of nutrition and dietetics, School of Public Health, College of Medicine and Health Sciences, Bahir Dar University, Bahir Dar, Ethiopia, **6** Department of Pharmacy, School of Health Sciences, College of Medicine and Health Sciences, Bahir Dar University, Bahir Dar, Ethiopia

o These authors contributed equally to this work.
‡ AT, AY, MT, HG and EA also contributed equally to this work.
* teshagerhylemarriam@gmail.com

**Data Availability Statement:** All relevant data are within the paper and its Supporting Information files.

## Abstract

### Background

The beneficial effect of the dietary practice is significant reduction in the risk of developing diabetes related complication. Dietary practice among type 2 diabetes is not well-implemented in Ethiopia. Up to now, in the nation, several primary observational studies have been done on dietary adherence level and its determinants among type 2 diabetes. However, a comprehensive review that would have a lot of strong evidence for designing intervention is lacking. So, this review with a meta-analysis was conducted to bridge this gap.

### Methods

A systematic review of an observational study is conducted following the PRISMA checklist. Three reviewers have been searched and extracted from the World Health Organization-Hinari portal (SCOPUS, African Index Medicus, and African Journals Online databases), PubMed, Google Scholar and EMBASE. Articles' quality was assessed using the Newcastle-Ottawa Scale by two independent reviewers, and only studies with low and moderate risk were included in the final analysis. The review presented the pooled proportion dietary adherence among type2 diabetes and the odds ratios of risk factors favor to dietary adherence after checking for heterogeneity and publication bias. The review has been registered in PROSPERO with protocol number CRD42020149475.

**Funding:** The authors received no specific funding for this work.

**Competing interests:** The authors have declared that no competing interests exist.

## Results

We included 19 primary studies (with 6, 308 participants) in this meta-analysis. The pooled proportion of dietary adherence in the type 2 diabetes population was 41.05% (95% CI: 34.86–47.24, $I^2$ = 93.1%). Educational level (Pooled Odds Ratio (POR): 3.29; 95%CI: 1.41–5.16; $I^2$ = 91.1%), monthly income (POR: 2.50; 95%CI: 1.41–3.52; $I^2$ = 0.0%), and who had dietary knowledge (POR: 2.19; 95%CI: 1.59–2.79; $I^2$ = 0.0%) were statistically significant factors of dietary adherence.

## Conclusion

The overall pooled proportion of dietary adherence among type 2 diabetes in Ethiopia was below half. Further works would be needed to improve dietary adherence in the type 2 diabetes population. So, factors that were identified might help to revise the plan set by the country, and further research might be required to health facility fidelity and dietary education according to diabetes recommended dietary guideline.

## Background

Half a billion people are living with diabetes worldwide and the number is projected to increase by 25% in 2030 and 51% in 2045 [1], particularly in low-and middle-income countries [2]. Global burden of type 2 diabetes is projected to increase to 7079 individuals per 100,000 by 2030, reflecting a continued rise across all regions of the world [3]. The goals in caring for patients with diabetes mellitus are to eliminate symptoms and to prevent, or at least slow, the development of complications [4].

Diabetes self-management is an essential component of effective self-care practice. Mindful eating offers promise as an effective approach for weight glycemic control in people with type 2 diabetes [5, 6]. Improvement in the elevated blood glucose level can be achieved through diet management [7]. Diets rich in whole grains, fruits, vegetables, legumes, and nuts; moderate in alcohol consumption; and lower in refined grains, red or processed meats, and sugar-sweetened beverages have been shown to reduce the risk of diabetes and improve glycemic control and blood lipids in patients with diabetes [8, 9].

Adherence to a healthy diet is the cornerstone in the prevention and management of diabetes [10, 11]. Medical nutrition therapy is an integral component of diabetes management and of diabetes self-management education [12]. Failure to follow a strict diet plan is leading causes of complications among patients with type 2 diabetes. There are many misconceptions exist concerning nutrition and diabetes, for example (a) do not allocate enough time for baste buds to change; (b) too impatient to see results; (c) having the wrong mindset; (d) lack of support from family, friends and support groups; and (e) lack preparation and planning [9, 13, 14].

The main factors to dietary adherence were both systemic (population changes, poor access to diet, cultural influences, and low-quality healthcare) and personal (poverty and cost, educational status, and perceptions about the disease) in nature [10]. The lack of proper professional dietary assessment, follow-up and advice by the health care providers are the main influence on dietary practice of type 2 diabetes [15].

Dietary adherence in type 2 diabetes has been found to vary from region to region in Ethiopia [16–21]. Even though the pooled proportion (50.18%) of good dietary practice among type 2 diabetes is documented in Ethiopia with small sample size [22], the overall dietary adherence

and common factors that promote good dietary practice are not documented in the country. Thus, this study aimed to assess the pooled proportion of dietary adherence and associated factors among type 2 diabetes population in Ethiopia.

## Materials and methods

### Protocol design and registration

A systematic review of an observational study was conducted following the meta-analysis of observational studies in an epidemiology statement. The Preferred Reporting Items for Systematic Reviews and Meta-Analyses Protocol (PRISMA-P) [23, 24]) and Meta-Analysis of Observational Studies in Epidemiology (MOOSE) guideline [25] were used for the development of this study protocol. The protocol of this systematic review and meta-analysis was registered with the International Registration of Systematic Reviews (PROSPERO) with PROSPERO registration number CRD42020149475. The protocol registration aimed to minimize duplication of the same reviews, provide transparency, and reduce reporting bias.

### Eligibility criteria

The studies (all published and unpublished) were that used observational epidemiological designs (cross-sectional), intended to assessed dietary adherence and associated factors among people with type 2 diabetes aged 15 and above, and articles that were published in the English language. Studies were conducted in type 1 diabetes and mixed both types, a case series, unclear definition of dietary adherence practice (like unclear measurement of questionnaires in the outcome variables, and did not report specific outcomes for dietary adherence/non-adherence quantitatively) were excluded in the final analysis.

### Data sources and searching strategy

A search strategy has been developed using fundamental concepts in the research question: "dietary adherence," "recommended dietary practice," "therapy adherence," "treatment adherence," "medication intake adherence," "medication compliance," "patient compliance," "diabetes mellitus," "type 2 diabetes," "diabetes," "patients," "clients," and "factors," "determinants," "influences," "risk factors," "predictors" and "Ethiopia". For each key concept, appropriate free-text words and Medical Subject Heading (MeSH) were used and combined using Boolean operators such as "AND' and "OR." This enabled the retrieval of relevant articles that might have used different synonyms for the same word. Notably, to fit the advanced PubMed database, the search was strategy applied (S1 Table).

A pretest of the search strategy by three authors was performed in PubMed, and the actual electronic search was done between 25 February and March 5, 2021. Three reviewers implemented the electronic search in the following electronic databases: PubMed, Embase, GOOGLE SCHOLAR, CINAHL, MEDLINE, and Hinari electronic databases. Hinari is the World Health Organization (WHO) database portal for low and middle income countries and includes Web of Science, SCOPUS, African Index Medicus (AIM), Cumulative Index to Nursing and Allied Health Literature (CINAHL), WHO's Institutional Repository for Information Sharing (IRIS), and African Journals Online databases. In addition, articles were also searched through a review of the grey literature available on institutional repositories (Addis Ababa, Bahir Dar, and Jimma Universities). Besides, we found other articles by snowballing and retrieving references lists of already identified articles to include studies that were unable to identify by search strategy.

## Study selection

All the citation identified by our search strategy, which was potentially eligible for inclusion, was exported to EndNote software version X7, Thomson Reuters, New York, NY, and the duplicate were removed. Title and abstracts of the remaining citation were screened by three independent reviewers (TWA, AT and AY) and ineligible studies were further excluded. The full texts of selected articles were retrieved and read thoroughly to ascertain their suitability before data extraction. The articles that fulfilled the earlier criteria have been used as sources of data for analysis.

## Data extraction

The abstract and full-text review and data abstraction were done by two independent reviewers (HG and MB) using a standardized data abstraction form, developed according to the sequence of variables required from primary studies on MS-Excel sheet. The disagreement between the two reviewers was resolved by a third independent reviewer through discussion (SA or EA). Before analysis, a transformation of the adjusted odds ratios and proportion was made.

The New castle Ottawa Scale criteria were selected for quality assessment of selected studies before analysis [26]. Two independent reviewers (TWA and MT) critically appraised each article using the NOS. Discrepancies between reviewers resolved by discussion and by including a third reviewer (EA). We used the average of two independent reviewer's quality scores used to decide whether the articles included or not. Articles with methodological flaws or incomplete reporting of results in the full-text excluded from the analysis. The data extraction formant included primary author, publication year, region, outcome measuring tool, study design, response rate, sample size, and proportion.

## Outcome measurement

The primary outcome of this review was the pooled proportion of dietary adherence among people with type 2 diabetes in Ethiopia. The proportion measured as the number of adult type 2diabetes with dietary adherence in the studies divided by the total number of type 2 diabetes people in a study multiplied by 100. For the analysis of the secondary outcomes (factors), we extracted data on factors that were related to dietary adherence practice in the literature, such as educational level, monthly income, knowledge towards diabetes and diabetes management, and received diabetes self-management education. In examining factors associated with dietary adherence, data used from the primary studies of the Adjusted Odd Ratios (AOR) to find the association between the independent variables and having dietary adherence practice.

## Quality assessment

The risk of bias of included studies is assessed using the 10-item rating scale developed by Hoy et al. for prevalence studies [27]. The assessment tool has the following domain of each articles; representative sample size, method of data collection, reliability and validity of study tools, case definition, and prevalence periods of the studies. Researchers categorized each study as having a low risk of bias ("yes" answers to domain questions) or a high risk of bias ("no" answers to domain questions). Each study assigned a score of 1 (Yes) or 0 (No) for each domain, and these domain scores were summed to give an overall study quality score. Scores of 8–10 were considered as having a "low risk of bias", 6–7 a "moderate risk", and 0–5 a "high risk" (S3 Table).

## Statistical analysis

**Testing for heterogeneity.** Heterogeneity between the results of the primary studies was assessed using Cochran's Q test and quantified with the $I^2$ statistics. A p-value of less than 0.1 was considered to suggest statically significant heterogeneity, considering a category a small number of studies and their heterogeneity in design [28]. Heterogeneity had taken low, moderate, and high categories when the $I^2$ values were below 25%, between 25% and 75%, and above 75%, respectively [29, 30]. Thus, the random effect model was used to pool the proportion of dietary adherence since the studies were found heterogeneous. The random effect model accounts for heterogeneity among study results beyond the variation associated with chance, unlike the fixed-effect model [31].

To investigate the source of heterogeneity, the random-effects meta-regression was conducted by taking primary study characteristics such as region, and study setting (types of hospitals) and outcome measurement tool. The meta-regression analysis was weighted to account for the residual between-study heterogeneity (i.e., heterogeneity not explained by the covariates in the regression) [32]. Subgroup analyses by region, type of study setup (types of hospitals) and outcome measurement tool were carried out because of significant heterogeneity between studies (i.e., $I^2$ = 93.1%, p < 0.05).

**Publication bias assessment.** Publication bias was assessed by visual inspection of funnel plots based on the shape of the graph (subjective assessment). The symmetrical graph was interpreted to suggest an absence of publication bias, whereas an asymmetrical one indicates the presence of publication bias. On the other hand, qualitatively (objective evaluation), Egger's weighted regression tests was used to assess publication bias with a p-value less than 0.1 considered as indicative of a statistically significant publication bias [33].

**Sensitivity analysis.** Lastly, a sensitivity analysis was done to estimate whether the pooled effect size was affected by single studies. A leave-one-out sensitivity analysis was performed to confirm whether there were studies that potentially biased the direction of the pooled estimate.

# Result

## Study selection

The database search and desk review yielded a total of 976 articles. Of these, 968 articles were retrieved from PubMed, Google Scholar, EMBASE, and the World Health Organization's Hinari portal (which includes the SCOPUS, African Index Medicus, and African Journals Online databases). The remaining 8 observational studies were found from institutional repositories (Addis Ababa, Gondar, and Bahir Dar Universities). After reviewing the titles and abstracts, we excluded 621 articles due to duplication. In screening, we excluded 332 articles because their outcomes were not in line with the desired eligibility criteria. The full-text of the remaining 22 articles has been accessed for eligibility and quality. Additionally, three articles were excluded because their outcomes variables were not clearly stated (34–36). The remaining 19 studies were included in the analysis (Fig 1).

## Study characteristics

Overall, a total of 19 observational studies were selected in this systematic review and meta-analysis. This consisted of 6308 participants (aged 15–85 years). The number of participants in each study ranged from 207 to 576. All studies were cross-sectional design to estimate dietary adherence. The most retrieved studies (n = 6) were from Oromia region [18, 37–41], followed by Addis Ababa (n = 4) [19, 20, 42, 43], and Amhara region (n = 4) [16, 17, 44, 45], SNNP [46, 47] and Dire Dawa [21, 48] were represented by two studies, whereas Tigray region region was

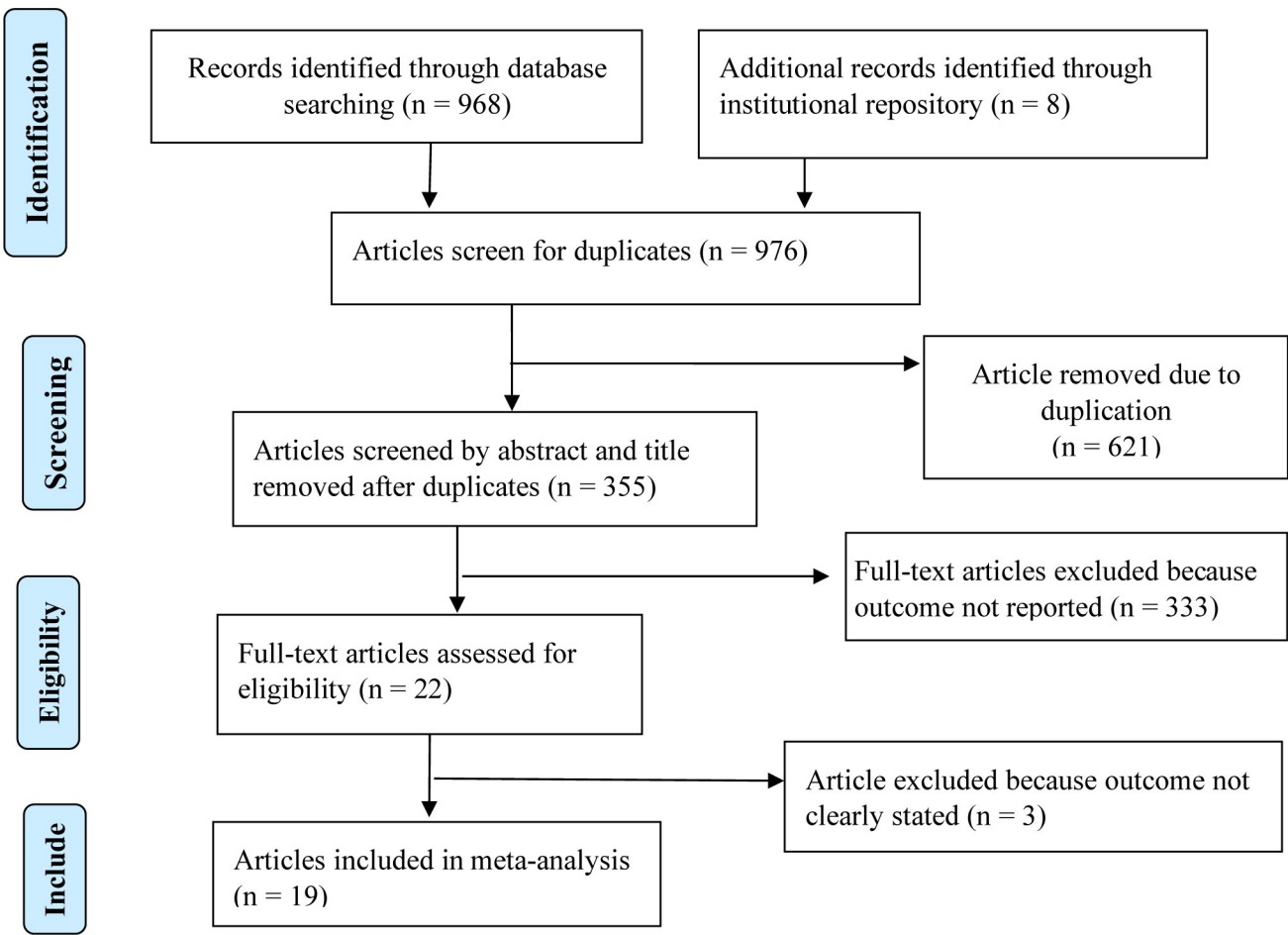

**Fig 1. PRISMA statement presentation for a meta-analysis of a pooled proportion of dietary adherence among type 2 DM in Ethiopia, 2012–2020.**

represented by one study [49]. Except for two studies [42, 48], all the studies have been reported in peer-reviewed journals. All of the studies have been reported high response rates (> 92.9%) (S2 Table, Table 1).

## Quality appraisal

The quality score of the included study ranged from 6 to 8 with a mean score of 8 .14 (SD = 0.91). Out of 19 studies, 13 (68.42%) studies received a low risk of bias, and four (21.05%) studies received a moderate risk of bias. The authors also find types of bias: eight studies [21, 41, 43–45, 47, 48] had a high risk of representation bias and seven studies [21, 37–39, 42, 43, 45] had a high risk of case definition bias (S3 Table).

## Meta-analysis

**Pooled estimates of dietary adherence among type 2 DM in Ethiopia.** The analysis of twenty observational studies was ranked as low and moderate-quality. The pooled proportion of dietary adherence of people with type 2 diabetes was 41.05% (95%CI: 34.86–47.24, $I^2$ = 93.1%). The highest (64%) [18] and the lowest (18.2%) dietary adherence reported in the Oromia region. High heterogeneity was observed among the included studies (Q test P<0.001) and $I^2$ ($I^2$ = 93.1%) (Fig 2). Due to the heterogeneity of included studies, further sub-group

**Table 1. Descriptive summary of 20 studies included in the meta-analysis of the proportion of dietary adherence among type 2 diabetes population in Ethiopia from 2012–2020.**

| Authors name | Study year | Source | Region | study tools | Hospitals | Age | RR | SS | Outcome | P | Quality score |
|---|---|---|---|---|---|---|---|---|---|---|---|
| Ayele AA et al | 2017 | Journal | Amhara | PDAQ | General | >18 | 100 | 320 | 82 | 26 | 8 |
| Berhe KK et al | 2012 | Journal | Tigray | SDSCA | Referral | >18 | 96.8 | 300 | 92 | 30.7 | 6 |
| Berhe KK et al | 2012 | Journal | A.A | SDSCA | Referral | > = 30 | 99.1 | 320 | 68 | 21 | 8 |
| Bonger Z et al | 2013 | Journal | A.A | Other | Referral | >18 | 100 | 419 | 101 | 24 | 8 |
| Daba A et al | 2020 | Journal | Oromia | Other | General | > = 30 | 100 | 248 | 45 | 18 | 6 |
| Degefa G et al | 2020 | Journal | SNNP | SDSCA | Referral | > = 30 | 95 | 207 | 113 | 55 | 7 |
| Demilew YM et al | 2016 | Journal | Amhara | Other | Referral | > = 40 | 94.8 | 401 | 144 | 36 | 8 |
| Fekadu et al | 2019 | Journal | Oromia | Other | Referral | > = 30 | 100 | 228 | 74 | 33 | 7 |
| Getie A et al | 2018 | Journal | Dire Dawa | Other | General | >18 | 98.6 | 506 | 228 | 45 | 7 |
| Halima MI et al | 2017 | Journal | Amhara | Other | Referral | > = 21 | 96.8 | 410 | 167 | 41 | 6 |
| Lemessa F et al | 2014 | IR | A.A | SDCA | Referral | >18 | 98.8 | 324 | 133 | 41 | 8 |
| Mekonnen et al | 2020 | Journal | Amhara | Other | Referral | > = 30 | 99.3 | 576 | 278 | 48 | 7 |
| Mohammed AS et al | 2019 | Journal | Dire Dawa | PDAQ | Referral | ≥18 | 100 | 307 | 115 | 38 | 8 |
| Rukiya D et al | 2018 | IR | Oromia | Other | Referral | > = 18 | 92.9 | 392 | 189 | 48 | 6 |
| Sorato MM et al | 2015 | Journal | SNNP | Other | General | > = 15 | 100 | 194 | 116 | 60 | 7 |
| Woldu MA et al | 2014 | Journal | Oromia | Other | General | >18 | 100 | 102 | 60 | 59 | 6 |
| Worku A et al | 2014 | Journal | A.A | MMAS-8 | Referral | ≥18 | 95.5 | 403 | 196 | 49 | 8 |
| Zeleke Negera G et al | 2019 | Journal | Oromia | SDSCA | Referral | ≥18 | 100 | 322 | 206 | 64 | 7 |
| Zinab B et al | 2018 | Journal | Oromia | MMAS-8 | General | > = 18 | 92.9 | 329 | 159 | 48 | 7 |

SS: Sample size; RR: Response Rate; IR: Institutional repository; A.A: Addis Ababa, NOS: New-castle Ottawa Scale; SNNP: Southern Nations Nationalities and People's; P: Prevalence; PDAQ: Perceived Dietary Adherence Questionnaire; MMAS: Morisky Medication Adherence Scale; SDSCA: Summary of Diabetes Self-Care Activities.

analysis was done by using the following study characteristics: regional location (Fig 2), outcome measurement tools and study setting (types of hospital). The random-effect model was applied for reporting the pooled proportion of dietary adherence of the sub-group analysis.

**Subgroup analysis.** On subgroup analysis by region, the highest pooled estimation of dietary adherence was found in the SNNP region (Pooled Proportion (PP) = 57.14%; 95% CI: 51.54, 62.75) and the lowest in the Addis Ababa (PP = 33.51%; 95% CI: 20.87, 46.30). The pooled estimation of dietary adherence among type 2 DM in the Dire Dawa city administration was 41.21% (95% CI: 33.76, 48.66), and 44.84% (95% CI: 30.19, 59.50) in Oromia. The pooled estimation of dietary adherence in a referral hospital is 40.40% (95% CI: 33.78, 47.02). The pooled estimation of dietary adherence was high in MMAS-8 scale measurement of dietary adherence level without heterogeneity 48.40% (95% CI: 43.02, 53.78) Table 2).

**Publication bias.** Both funnels plots of precision asymmetry and the Egger's test of the intercept showed that there is no publication bias in the primary studies. Visual examination of the funnel plot showed a symmetric distribution of studies. Additionally, Egger's test of the intercept was 0.291 (95% CI: -0.085, 0.667) p > 0.05 (0.121), as judged by Egger's test. This is suggesting that publication bias estimates were not statistically significant (Fig 3).

**Meta-regression and sensitivity analysis.** The sub-group analysis showed that heterogeneity across the studies was widespread. To name the source of heterogeneity, we conducted a meta-regression and sensitivity analysis. During the meta-regression analysis, we conducted using the following study covariance: study years, sample size, and region. However, the results showed that none of these variables were a statistically significant source of heterogeneity (Table 3).

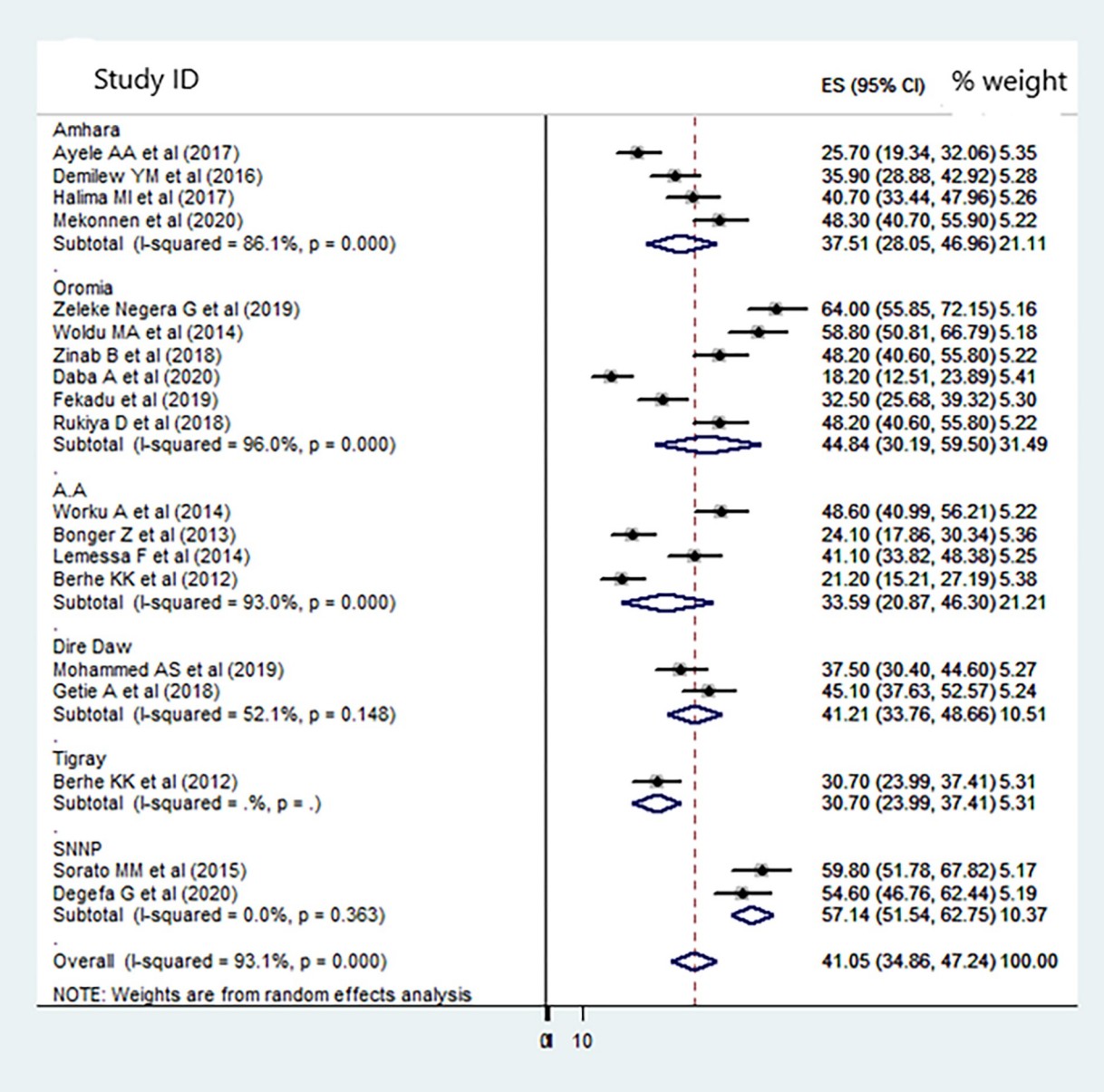

**Fig 2. A subgroup analysis of the forest plot showing the pooled proportion of dietary adherence among type 2 DM in Ethiopia, 2012–2020.**

We also performed a sensitivity analysis to find the influence of each study on the overall effect size. No single primary study affected the overall pooled proportion of dietary adherence among people with type 2 diabetes in Ethiopia (Fig 4).

## Determinants of dietary adherence

Extracted adjusted odds ratios from the primary studies were educational level, average monthly income, and dietary knowledge and pooled to identify predominantly associated factors for dietary adherence. Accordingly, people who had high level of education (Pooled Odds Ratio (POR): 3.29; 95%CI: 1.41–5.16; $I^2 = 91.1\%$) (Fig 5), people who had high level of average monthly income (POR: 2.50; 95%CI: 1.41–3.52; $I^2 = 0.0\%$) (Fig 6), and those who had dietary

**Table 2. Sub-group analysis of dietary adherence based region, hospital and outcome measurement tool in Ethiopia from 2012 to 2020.**

| Variables | Characteristics | Estimated proportion of dietary adherence (95%CI) | I²% (p value) |
|---|---|---|---|
| Region | Tigray | Single study | Single study |
| | Oromia region | 44.84 (30.19, 59.50) | 96.0 (<0.001) |
| | Dire Dawa | 41.21 (33.76, 48.66) | 52.1 (>0.001) |
| | Addis Ababa | 33.51 (20.87, 46.30) | 93.0 (<0.001) |
| | Amhara | 37.51 (51.54, 62.75) | 86.0 (<0.001) |
| | SNNP | 57.14 (51.54, 62.75) | 0.0 (>0.001) |
| Outcome measurement tool | | | |
| | SDSCA | 42.49 (22.96, 62.01) | 96.7 (<0.001) |
| | MMAS-8 | 48.40 (43.02, 53.78) | 0.0 (0.942) |
| | PDAQ | 31.49 (19.93, 43.05) | 83.0 (0.015) |
| | Others* | 41.0 (32.14, 49) | 93.6 (<0.001) |
| Hospitals | General | 42.48 (28.01, 56.96) | 96.0 (<0.001) |
| | Referral | 40.40 (33.78, 47.02) | 91.1 (<0.001) |

SNNP: Southern Nations Nationalities and People's; PDAQ: Perceived Dietary Adherence Questionnaire; MMAS: Morisky Medication Adherence Scale; SDSCA: Summary of Diabetes Self-Care Activities; *outcome measurement not measure one of PDAQ, MMAS or SDSCA.

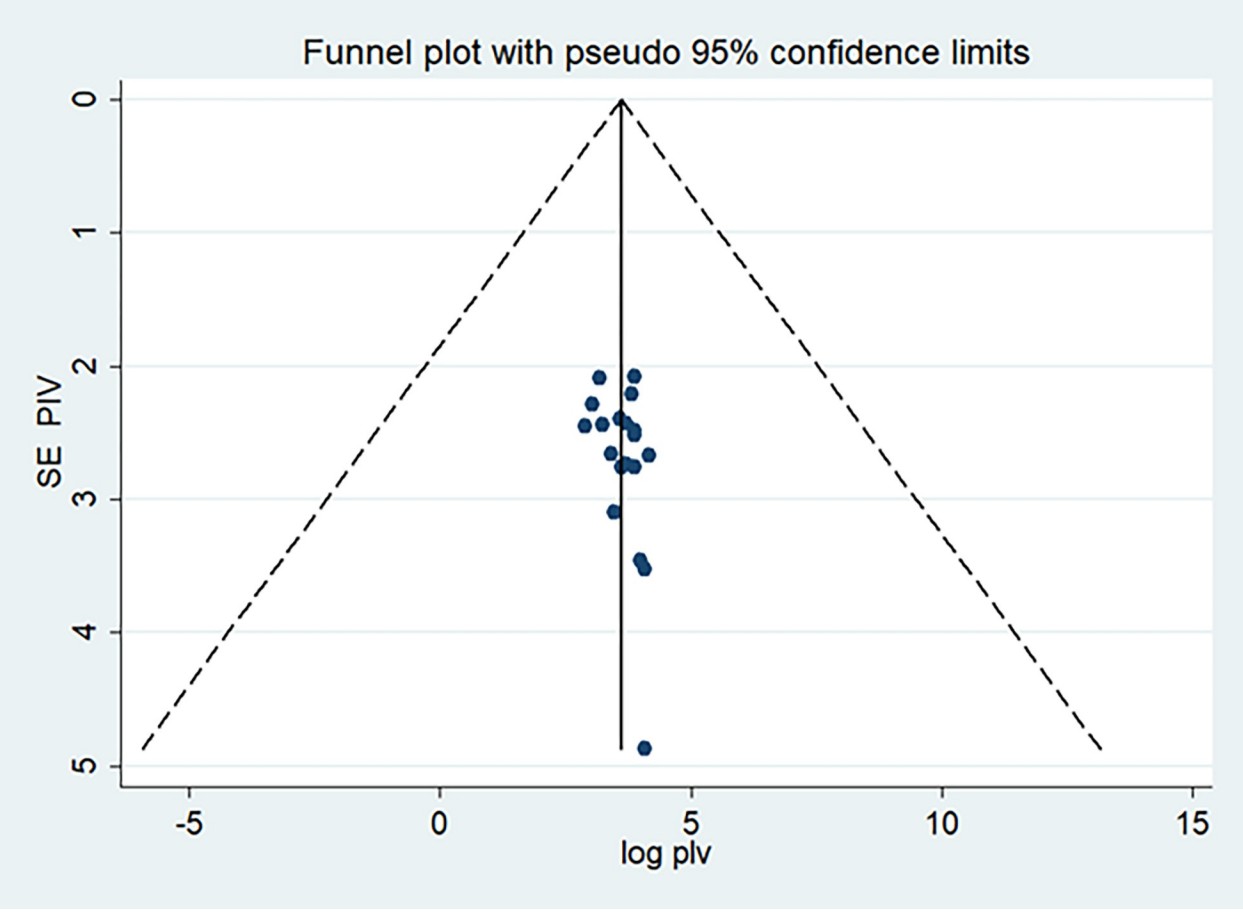

**Fig 3. Meta funnels presentations of the proportion of dietary adherence among type 2 DM in Ethiopia, 2012–2020, whereby SE PIV (standard error of proportion) plotted on the Y-axis and log PIV (logarithm of proportion) on the X-axis.**

**Table 3. Meta-regression output to explore heterogeneity of the pooled proportion of dietary adherence among type 3 diabetes population in Ethiopia from 2012–2020.**

| Variables | Coefficients | P-value | 95% CI |
|---|---|---|---|
| Study year | 0. 38 | 0.880 | -4.93, 5.70 |
| Sample size | 0. 090 | 0.791 | - 0.166, 0.21 |
| Region | | | |
| Tigray | Single study | | Single study |
| Oromia region | 14.08 | 0.343 | -16.84, 45.00 |
| Dire Dawa | 10.63 | 0.523 | -24.36, 45.62 |
| Addis Ababa | 2.99 | 0.843 | -28.95, 34.94 |
| Amhara | 6.97 | 0.645 | -24.96, 38.91 |
| SNNP | 26.49 | 0.128 | -8.69, 61.68 |

knowledge (POR: 2.19; 95%CI: 1.59–2.79; $I^2$ = 0.0%) (Fig 7) are determinate factors to adhere diet.

## Discussion

To the best of our review, this is the first systemic review and meta-analysis study that conducted to show the pooled proportion of dietary adherence and associated factors among type 2 diabetes in Ethiopian. This study identified that less than half people with type 2 diabetes (41.05%) in Ethiopia adhere to dietary practice as recommended or agreed between people with type 2 diabetes and healthcare providers or standard dietary recommendation guideline.

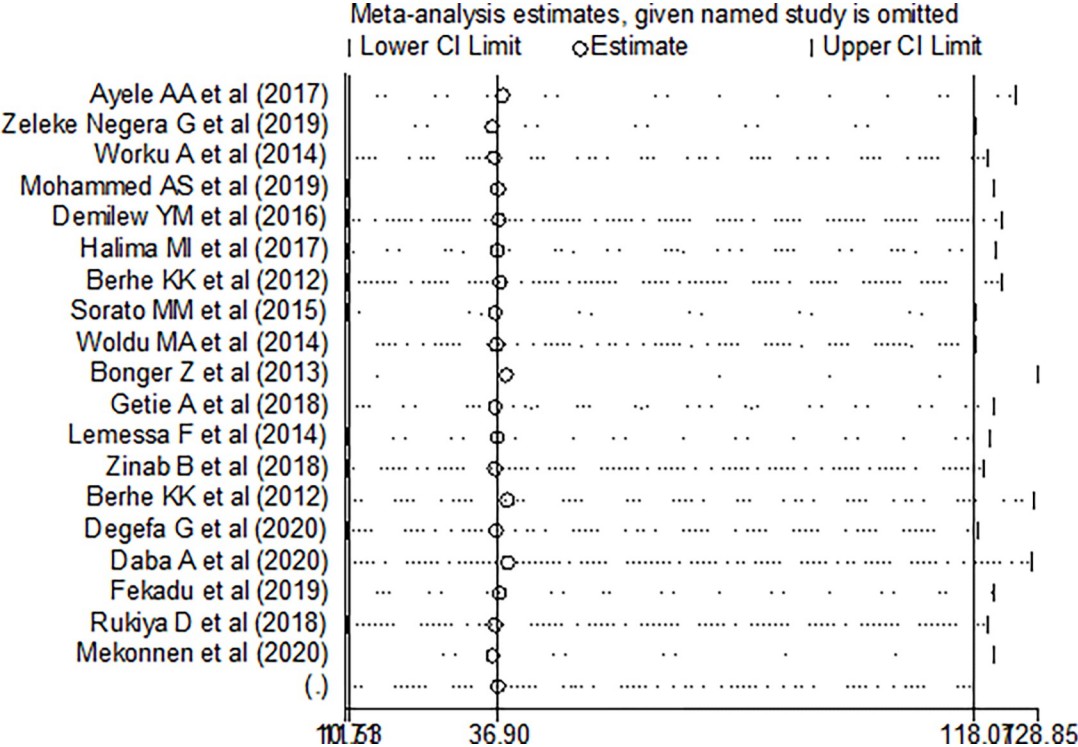

**Fig 4. One-leave-out sensitivity analysis for studies conducted on proportion of dietary adherence among people with type 2 DM in Ethiopia, 2012–2020.**

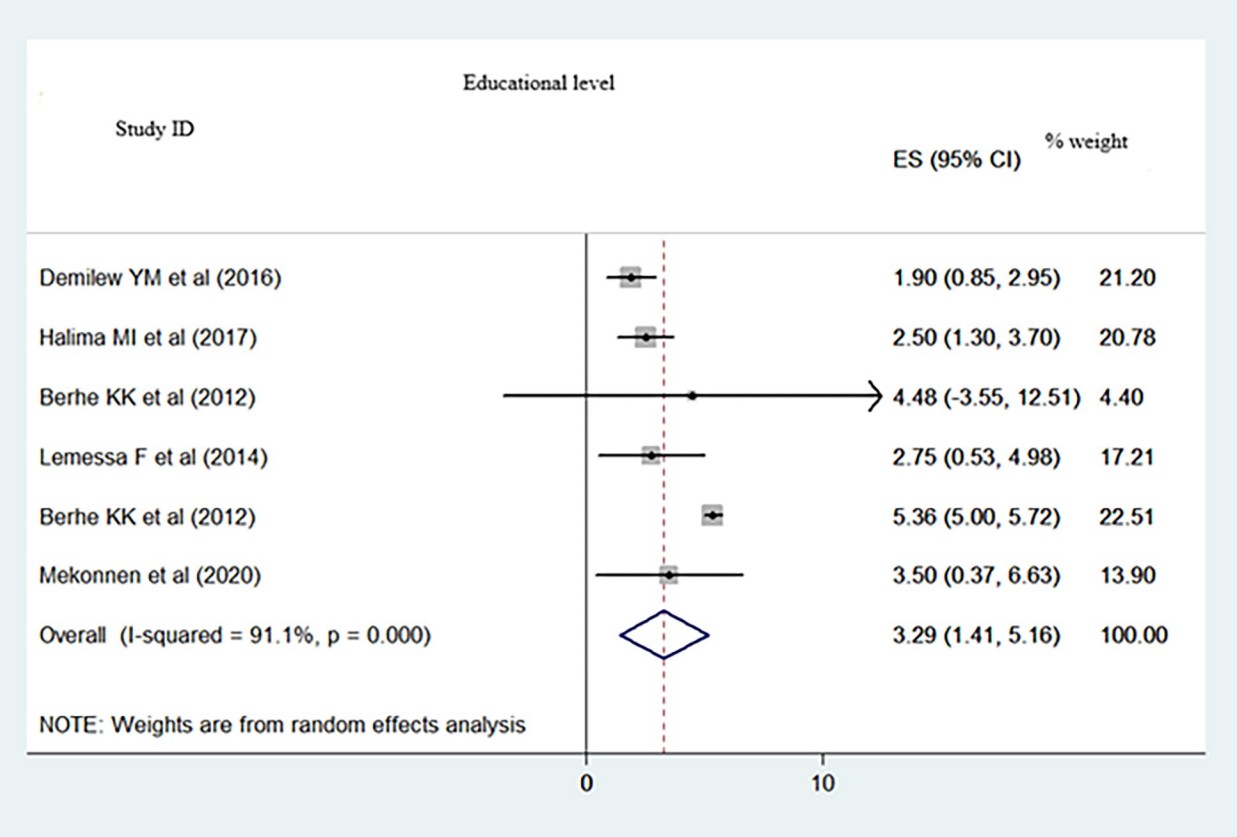

**Fig 5. A meta-analysis of educational level associated with dietary adherence among type 2 DM population in Ethiopia.**

This finding is higher than a study conducted in Brazil (29.2%) [50, 51], Bangladeshi (22%) [52]; but lower than a study conducted in Iran (94.6%) [53]. This is because, in Brazil, consumption of unhealthy diet markers was greatest in diabetes population [54]. Unhealthy eating habits among type 2 diabetes are a significant impact in Bangladeshi [55]. A qualitative study in Iran documented that the food culture largely affects a type 2 DM patient's adherence to a healthy diet [56]. In Ethiopia, the probable because of (a) low dietary adherence is economic burden of diabetes care is very disastrous among the less privileged populations group [57] (b) no access of healthy dietary habit [7, 58] (c); and not access of recommended diet [59, 60].

Dietary adherence emphasizes the importance of minimizing macro-vascular and micro-vascular complications in people with diabetes [61]. A recommended diet that improve metabolic conditions for type2 DM are Mediterranean diet, a low-carbohydrate/high-protein diet, a vegan diet and a vegetarian diet [61, 62]. That why good dietary adherence improves the effectiveness of pharmacological intervention and promoting healthy lifestyles [63].

People who had high level of education (Pooled Odds Ratio (POR): 3.29; 95%CI: 1.41–5.16; $I^2$ = 91.1%) (Fig 5), people who had high level of average monthly income (POR: 2.50; 95%CI: 1.41–3.52; $I^2$ = 0.0%) (Fig 6), and those who had dietary knowledge (POR: 2.19; 95%CI: 1.59–2.79; $I^2$ = 0.0%) (Fig 7) were determinate factors to adhere diet.

The educational level is a potential determinate to dietary adherence among type 2 DM individuals. Those people with type 2 diabetes who have high levels of education had more likely to dietary adherence behaviors. This finding is in line with a meta-analysis done in China [63], in a large-scale cross-sectional study in Switzerland [64], and Bangladesh [52].

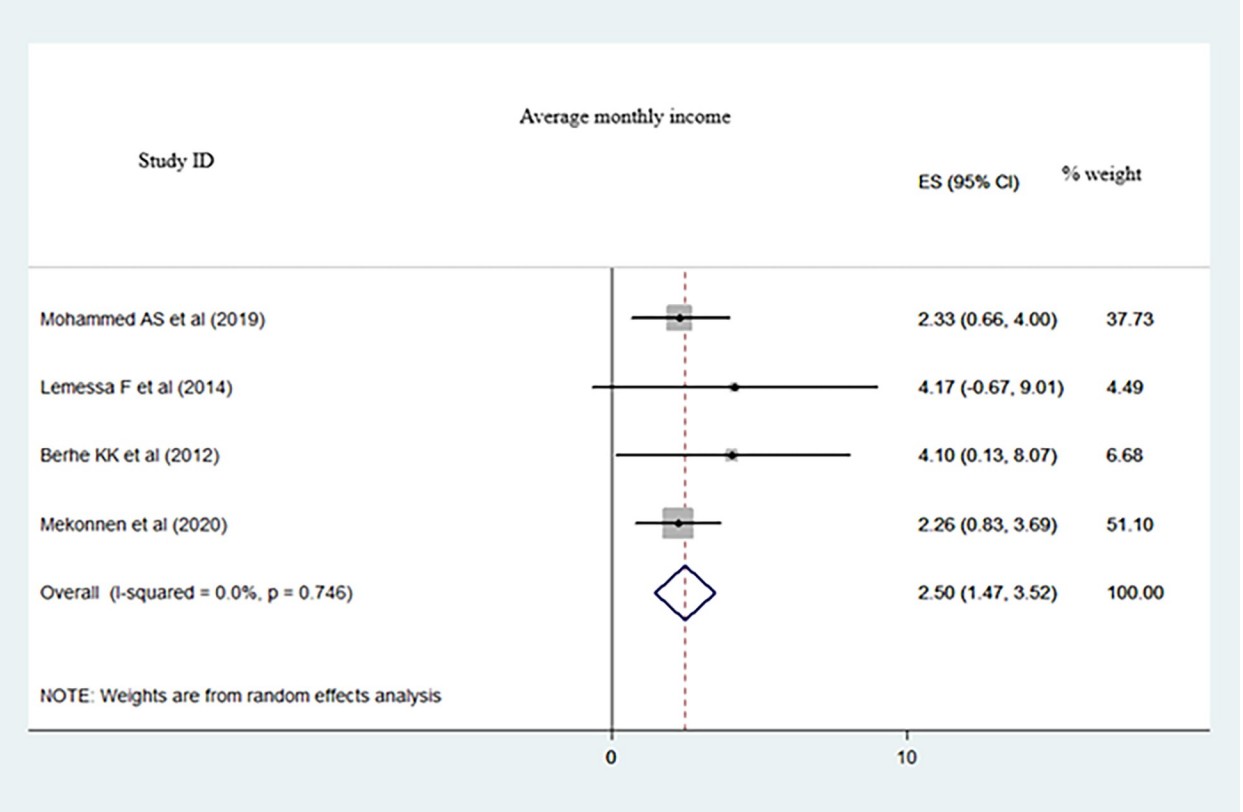

**Fig 6. A meta-analysis of average monthly income associated with dietary adherence among type 2 DM population in Ethiopia.**

Evidence shows that when people were more educated improves people's dietary adherence [65], and low educated level linked with inadequate glycemic controls [66]. Scholars identified that people with lower levels of education consume sugar- and fat-rich foods more often and fruit and vegetables less often than adults with a high education level [67]. Therefore, educational level modulates the level of dietary adherence among study participants. Higher education may be related to knowledge and awareness of healthy eating habits. Being higher education levels could lead to having better judgment and decision-making ability for choosing healthy diet and eating behaviors [67].

Participants' monthly income is associated with dietary adherence. This finding in line with the former studies [68, 69] significantly positive association had observed between higher average monthly income level and better dietary practice. Because participants who had higher income did not worry about food choices, had no difficulty resisting the temptation to eat unhealthy food, and afford healthy food even if being too expensive [68]. Therefore, low-income participants are prone to follow unhealthy diet [69].

The following reason may explain income differences in dietary adherence among type 2 diabetes. (a) High cost of healthy foods, (b) healthy diet may be perceived as being expensive in comparison unhealthy diet group [70, 71], (c) the expenditure of healthy diets increased with increasing income quintiles [72], and (d) socio-economic disparities among individuals [73].

People with type 2 diabetes who have had moderate and high level of dietary knowledge had double pronounced dietary adherence than people with type 2 diabetes who have had a low level of dietary knowledge. This finding agrees with the earlier study [74, 75]. Dietary

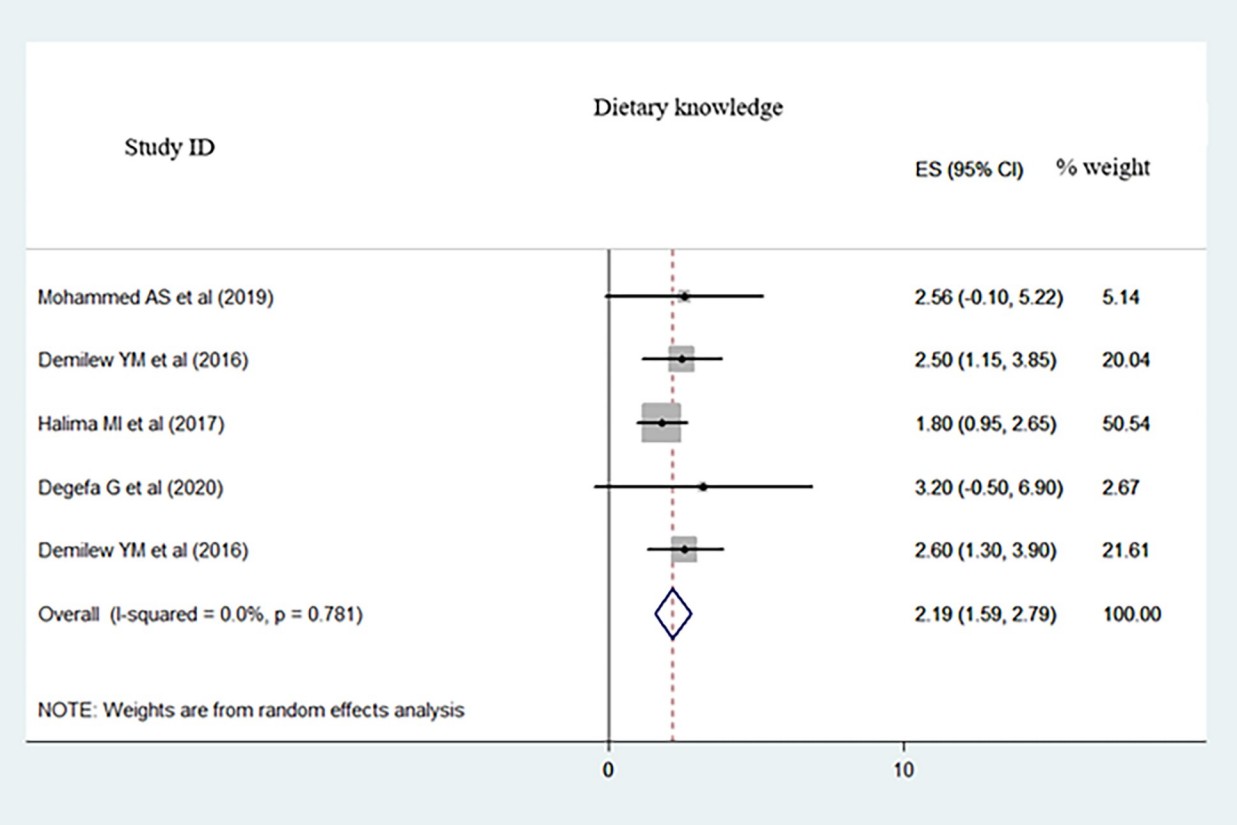

**Fig 7. A meta-analysis of dietary knowledge associated with dietary adherence among type 2 DM population in Ethiopia.**

knowledge enable individuals with type 2 diabetes to make food choices that (a) optimize metabolic self-management, (b) prevent complication, (c) reduced or hold drug intake, and (d) improve quality of life [74].

In addition to this, dietary knowledge is related to self-care behaviors and good dietary adherence [76]. That is why positive aspects of dietary knowledge are healthy eating like increased fruit and vegetable consumption; and a low intake of simple sugars, fat, and salt. Awareness of food, nutritional properties, and recommendations on the size and frequency of consumption must be the primary goals of nutritional education programmers [77]. Dietary knowledge is an integral component of health literacy [78]. Dietary knowledge (high health literacy) of chronic disease patients like type 2 DM is associated with low healthcare costs, optimal self-management, and good health outcomes, especially for type 2 diabetes [79]. Dietary knowledge helps individuals' impression, process, understand, and communicate diet-related information needed to make informed health decisions [80].

## Strength and limitations

This systematic review and meta-analysis have some strength. This study pooled several studies that provide evidence of the pooled proportion of dietary adherence and its determinant among people with type 2 DM. It includes a large sample size which is much more than the sample sizes of each study. We tried to pool the estimated pooled proportion of common determinate of dietary adherence in the nation.

Despite its strengths, the study also has a few limitations. Even though most of the studies had good quality, all primary studies incorporated in this meta-analysis were cross-sectional which is limited in this study. Besides, we tried extensive and diverse search strategies to find all possible available literature, some grey literature, such as conference proceedings, remained difficult to find which turned limit this study. Furthermore, methods applied to measure outcome parameters varied among included studies.

## Implication

This study has many implications for clinical practice and future research. Firstly, the health care provider (especially physicians, nurses, dietician, and health educators) can develop effective strategies (follow dietary plan, dietary education and tailoring dietary interventions to a person's dietary preferences) to improve dietary adherence. Secondly, identifying and understanding factors that favor and restrict dietary practice is the second step in developing evidence-based interventions to promote short and long-term health outcomes and quality of life. Future research should focus on developing and testing a conceptual model (collaborative, client empowerment and enhance capacity link them to dietician and community resource) that can use to enhance dietary adherence in a national context. Ensure that the recommended diets (food) are availability, affordability, and cultural acceptability in the context of Ethiopians.

## Conclusions

This meta-analysis revealed that a low proportion of dietary adherence among people with type 2 DM. Educational level, monthly income, and dietary knowledge were significantly associated factors with dietary adherence. Therefore, health care personnel should build the type 2 DM clients' dietary knowledge based on what they see, hear, feel, and perceive the healthy diet. Increase awareness of the importance of healthy food habits is the first step in altering eating behavior. Ensure more focusing on eating a healthy diet is as preventive and curative measures for diabetes in health education programs as well in medical curricula.

## Supporting information

**S1 Checklist. PRISMA 2009 checklist.**
(DOC)

**S1 Table. Search strategy applied to PubMed database in the current review.**
(DOCX)

**S2 Table. Scoring of the quality of articles by authors using the Newcastle-Ottawa quality assessment tool.**
(DOCX)

**S3 Table. Risk of bias assessment tool of eligible articles by using the Hoy 2012 tool.**
(DOCX)

## Author Contributions

**Conceptualization:** Teshager Weldegiorgis Abate, Minale Tareke, Selam Abate, Mulat Tirfie, Haileyesus Gedamu, Emiru Ayalew.

**Data curation:** Teshager Weldegiorgis Abate, Minale Tareke, Selam Abate, Abebu Tegenaw, Minyichil Birhanu, Mulat Tirfie, Ashenafi Genanew, Haileyesus Gedamu, Emiru Ayalew.

**Formal analysis:** Teshager Weldegiorgis Abate, Minyichil Birhanu, Alemshet Yirga, Ashenafi Genanew.

**Investigation:** Ashenafi Genanew.

**Methodology:** Teshager Weldegiorgis Abate, Alemshet Yirga, Ashenafi Genanew.

**Software:** Teshager Weldegiorgis Abate.

**Visualization:** Abebu Tegenaw, Haileyesus Gedamu, Emiru Ayalew.

**Writing – original draft:** Teshager Weldegiorgis Abate, Selam Abate, Abebu Tegenaw, Minyichil Birhanu, Mulat Tirfie, Ashenafi Genanew, Haileyesus Gedamu, Emiru Ayalew.

**Writing – review & editing:** Teshager Weldegiorgis Abate, Minale Tareke, Minyichil Birhanu, Alemshet Yirga, Mulat Tirfie, Ashenafi Genanew, Haileyesus Gedamu, Emiru Ayalew.

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
