## [Decision Letter · Decision Letter 0]

11 Feb 2022

PONE-D-21-20444

Dietary adherence and its determinants among type 2 diabetes population in Ethiopian: a systemic review with meta-analysis

PLOS ONE

Dear Dr. Abate,

Thank you for submitting your manuscript to PLOS ONE. After careful consideration, we feel that it has merit but does not fully meet PLOS ONE’s publication criteria as it currently stands. Therefore, we invite you to submit a revised version of the manuscript that addresses the points raised during the review process.

Specifically the reviews were positive, and the study is thought to be impactful. There are a number of suggested edits to the manuscript that may make the manuscript easier to read. In addition, there are a few questions that the reviewer has raised that should be addressed, including why the participants aged 30 and above were considered an inclusion criteria, and how you assured the third independent review’ quality.

We look forward to receiving your revised manuscript.

Kind regards,

Colin Johnson, Ph.D.

Academic Editor

PLOS ONE

Journal Requirements:

2. PLOS ONE does not copy edit accepted manuscripts. Please proofread for typos and grammar, for instance in the title.

3. Thank you for stating the following financial disclosure: "No. The funders had no role in study design, data collection and analysis, decision to publish, or preparation of the manuscript."

6. We noticed you have some minor occurrence of overlapping text with the following previous publication(s), which needs to be addressed:

- https://journals.plos.org/plosone/article?id=10.1371%2Fjournal.pone.0245862

In your revision ensure you cite all your sources (including your own works), and quote or rephrase any duplicated text outside the methods section. Further consideration is dependent on these concerns being addressed.

Reviewers' comments:

Reviewer's Responses to Questions

**Comments to the Author**

1. Is the manuscript technically sound, and do the data support the conclusions?

Reviewer #1: Partly

2. Has the statistical analysis been performed appropriately and rigorously? 

Reviewer #1: Yes

3. Have the authors made all data underlying the findings in their manuscript fully available?

Reviewer #1: Yes

4. Is the manuscript presented in an intelligible fashion and written in standard English?

Reviewer #1: Yes

5. Review Comments to the Author

Reviewer #1: Please see the attached file for the feedback and comments. Overall, the study has a significant impact on the improvement strategies of DM control in Ethiopia. However, some revisions are desirable to make this manuscript more persuasive. Proofreading and appropriate reference style are necessary to ensure the manuscript quality. There were some minor mistakes in the manuscript.

6. PLOS authors have the option to publish the peer review history of their article (what does this mean?). If published, this will include your full peer review and any attached files.

Reviewer #1: No

---

## [Author Response · Author response to Decision Letter 0]

15 Mar 2022

Response to Reviewers

Response to editor and reviewers’

Response to the editor: 

We thank you and the reviewers for a thorough reading and constructive criticism of our manuscript and for the opportunity to revise and resubmit. We are pleased to submit the improved research article, including a proposed comment, “Dietary adherence and its determinants among type 2 diabetes population in Ethiopian: a systemic review with meta-analysis with a manuscript ID of PONE-D-21-20444” 

1. General Comments:

#1. COMMENT: Please ensure that your manuscript meets PLOS ONE's style requirements, including those for file naming.

RESPONSE: We have checked and attest that all formatting and style requirements have

been met PLOS ONE's style requirements. 

#2. PLOS ONE does not copy edit accepted manuscripts. Please proofread for typos and grammar, for instance in the title.

RESPONSE: we try to proofread and avoid copy edited in the manuscript. 

#3. COMMENT: The funders had no role in study design, data collection and analysis, decision to publish, or preparation of the manuscript. At this time, please address the following queries…

RESPONSE: The authors received no specific funding for this work.

#4. COMMENT: In your Data Availability statement, you have not specified where the minimal data set underlying the results described in your manuscript can be found.

RESPONSE: all data available in the manuscript. We stated in submission system.

#5. COMMENT: Please include captions for your Supporting Information files at the end of your manuscript, and update any in-text citations to match accordingly.

RESPONSE: We include all capitation 

#6. COMMENT: We noticed you have some minor occurrence of overlapping text with the following previous publication(s), which needs to be addressed 

RESPONSE: We revise the manuscript to rephrase the duplicated and overlapping text.

2. Review Comments to the Author

REVIEWER #1 COMMENTS

1. COMMENT: Were the “independent reviewers” enough experienced to review? They should be

well-experienced reviewers, so it would be better to mention their level.

RESPONSE: they review each article independently. The have experience to review the quality of articles using assessment check list. The also published article systemic review and meta-analysis. They are assistant professors. 

2. COMMENT: Change “Is” to “was”. Please proofread the manuscript again to revise minor

mistakes.

RESPONSE: Thank you this comment. We change the word ‘is’ to ‘was’ in the revised manuscript. 

3. COMMENT: Remove one “with”. There are two “with”. 

RESPONSE: we remove the word ‘with’ which was a typo error. 

4. COMMENT: Why the participants aged 30 and above were considered an inclusion criteria? This is not consistent with the analyzed study target generation.

RESPONSE: We accept the constructive comment. We made editorial error in the inclusion criteria. We corrected as age 15 and above in the revised manuscript. 

5. COMMENT: “find articles by reviewing the 5 reference lists of already identified researches.” This sentence is not easy to understand. Please rephrase.

RESPOSE: we try to rephrase these sentence 

6. COMMENT: How did you assure the third independent review’ quality?

RESPOSE: through discussion using article review check list 

7. COMMENT: How did you assure the “Discrepancies between reviewers resolved by discussion”? How was the discussion conducted?

RESPONSE: through discussion using article review standard check list 

8. COMMENT: Please rephrase “the risk of bias of included studies”. It is difficult to follow.

RESPONSE: we try to clarify by rephrasing. 

9. COMMENT: “For 2 the least risk of bias classification, discrepancies between the reviewers resolved via consensus” How did you assure the validity? Did you use any references? ….

RESPONSE: we use standard reference and quality assessment check list as we cited and provide supporting data. The last sentence was mistakenly written and we remove it. 

10. COMMENT: “study setting (hospitals)” is this type of hospital? Please rephrase to make it clear.

RESPONSE: yes it is type of hospital (referral and general hospitals). We also include the manuscript. 

11. COMMENT: Why was P-value set at 0.1, not 0.05? P-value of 0.05 has more significant than that of 0.1 statistically

RESPONSE: As far as our knowledge, we considered P-value less than 0.1 is more significant in Egger’s test of regression. If we set P-value at 0.05, we increase a chance to include a small study which turn increase bias. We are ready to learn if we understood in the wrong way, thank you. 

12. COMMENT: “Eff ect” and “aff ected” It looks there are redundant space between f and e.

RESPONSE: we avoid space and rewrite again in the revised manuscript. 

13. COMMENT: The response rate was quite high. Any bias? Did all studies take appropriate ethical consideration measure?

RESPONSE: During our review of the study articles, we thoroughly review the quality of study including the ethical review for each articles. 

14. COMMENT: “study setting (hospital)” Same comment to page 8 line 14. 

RESPONSE: we rewrite and make consistence 

15. COMMENT: Miss typo “estimatio”

RESPONSE: Yes positively it is a miss typo, we corrected it. 

16. COMMENT: Please rephrase “Egger’s test of the intercept 5 was 0.291 (95% CI: -0.085, 0.667) p > 0.05 (0.121) as judged by Egger’s test”. It is not easy to follow.

RESPONSE: we try to rephrase and try to clear for reader.

17. COMMENT: What is the evidence of “Dietary adherence emphasizes the importance of minimizing macro-vascular and micro- vascular complications in people with diabetes?” 

RESPONSE: reference 62 is an evidence of the importance of minimizing macro-vascular and micro-vascular complication. 

18. COMMENT: What is these “interventions”? There are various interventions for DM control

RESPONSE: intervention stands for ‘pharmacological intervention’

19. COMMENT: The knowledge of healthy diet behaviour would be different from the knowledge

gained at primary/junior high schools. Why we can say this relation?

RESPONSE: Being higher education levels could lead to having better judgment and decision-making ability for choosing healthy diet and eating behaviors

20. COMMENT: It should be that low-income participants are prone to follow unhealthy diet. It is not consistent with the findings. Why "emotional distress" is raised here suddenly? What is the meaning of "emotional distress" here? 

RESPONSE: Yes it is not consistence. We re-edit as recommended.

21. COMMENT: “the cost of healthy diets increased with increasing 6 income quintiles” This is not clear. The expenditure of healthy diets?

RESPONSE: Accept the comment and re-edit in the revised manuscript. 

12. COMMENT: What is the level of knowledge compared to a low level of it?

RESPONSE: Moderate and high level of knowledge 

22. COMMENT: What is the difference of simple sugar and "sugar"? If there is no difference, it would be better to rephrase "sugar"

RESPONSE: we rephrase it. 

23. COMMENT: Why "nutrition knowledge" is started to discuss here. The main theme should be summarized at the last paragraph. “nutritional knowledge” should not be fully equal to dietary adherence and healthy diet behavior for type 2 DM.

RESPONSE: We rephrase as dietary knowledge and rearrange the paragraph in the revised manuscript. This is not the main theme rather it is a discussion of the implication of dietary knowledge to dietary adherence. 

24. COMMENT: What is the several strengths? I can identify only one strength of the sample size. Please mention clearly several strengths, if you want to say them here.

RESPONSE: we try to mention 1) large sample size, 2) pooled the proportion of dietary adherence and 3) pooled the proportion of common determinants of dietary adherence in the nation. As the review comment we rephrase the word several. 

25. COMMONT: What is the result of the trial? Just "tried" cannot be the strength of this study.

RESPONSE: we try to rephrase this sentence in the revised manuscript. 

26. COMMENT: Please rephrase “all primary studies were cross-sectional that is limited in this

study”. It is not easy to follow. 

RESPONSE: since we are not incorporated a primary study those with a method of case-report study, qualitative study, and longitudinal study design. 

Future research should focus on developing and testing a conceptual model that can use to enhance diabetes self-care practice in a national context. Finally, to give a long-term reduction in diabetes-related co-morbidity and mortality, researches should assess ways to extend and sustain diabetes self-care practice among this population.

27. COMMENT: What is the results of your trial? As the result, what is the limitation? Please rephrase to make clear the limitation. “Besides, we 8 tried extensive and diverse search strategies to find all possible available literature, some grey 9 literature, such as conference proceedings, remained difficult to find.”

RESPONSE: We try to rephrase these comment. 

28. COMMENT: What kind of “effective strategies” could be suggested from this study? Please

describe.

RESPONSE: We try to describe in bracket in the revised manuscript. 

29. COMMENT: “Future research should focus on development” What for is “development”?

RESPONSE: This is an editorial error. We complete the sentence and make clear the reader. 

30. COMMENT: “Testing recommends food availability, affordability, and cultural acceptability in the 18 context of Ethiopians.” Please rephrase. What is “testing”?

RESPONSE: rephrase the word testing in the revised manuscript

31. COMMENT: What is the evidence of “based on what they see, hear, feel, and perceive the

healthy diet”?

RESPONSE: Since it make unclear for reader, we rephrase as ‘and attitude towards healthy diet during dietary counseling and education’

32. COMMENT: Please rephrase “Ensure more focusing on eating a healthy diet is as preventive and curative measures for 3 diabetes in health education programs as well in medical

curricula.” to make the meaning clear.

RESPONSE: Rephrase the sentence to make clear reader

COMMENT: The style of Reference does not follow the guideline. Please recheck the style.

RESPONSE: We try to follow the guideline.

---

## [Decision Letter · Decision Letter 1]

30 Jun 2022

Level of dietary adherence and determinants among type 2 diabetes population in Ethiopian: a systemic review with meta-analysis

PONE-D-21-20444R1

Dear Dr. Abate,

  I have reviewed the changes you have made in the revised manuscript and differ with the reviewer in concluding that you have addressed the reviews sufficiently to warrant publication. I am therefore offering acceptance of your manuscript. 

Kind regards,

Colin Johnson, Ph.D.

Academic Editor

PLOS ONE

Additional Editor Comments (optional):

Reviewers' comments:

Reviewer's Responses to Questions

**Comments to the Author**

1. If the authors have adequately addressed your comments raised in a previous round of review and you feel that this manuscript is now acceptable for publication, you may indicate that here to bypass the “Comments to the Author” section, enter your conflict of interest statement in the “Confidential to Editor” section, and submit your "Accept" recommendation.

Reviewer #1: (No Response)

2. Is the manuscript technically sound, and do the data support the conclusions?

Reviewer #1: Partly

3. Has the statistical analysis been performed appropriately and rigorously? 

Reviewer #1: Yes

4. Have the authors made all data underlying the findings in their manuscript fully available?

Reviewer #1: Yes

5. Is the manuscript presented in an intelligible fashion and written in standard English?

Reviewer #1: No

6. Review Comments to the Author

Reviewer #1: The resubmitted manuscript was not revised following reviewer's comments, although the author responded to it that "Manuscript was revised". Was the resubmitted manuscript was the same one which submitted initially? If the author did not revise the manuscript, I cannot recommend to accept it. Inconsistency between the response and revision of the manuscript prevent me from reviewing it again. Please address all my previous comments and revise the manuscript certainly, not just responding to the comments, before the submission.

7. PLOS authors have the option to publish the peer review history of their article (what does this mean?). If published, this will include your full peer review and any attached files.

Reviewer #1: No

---

## [Editor Report · Acceptance letter]

29 Jul 2022

PONE-D-21-20444R1 

Level of dietary adherence and determinants among type 2 diabetes population in Ethiopian: a systemic review with meta-analysis 

Dear Dr. Abate:

I'm pleased to inform you that your manuscript has been deemed suitable for publication in PLOS ONE. Congratulations! Your manuscript is now with our production department. 

Kind regards, 

on behalf of

Dr. Colin Johnson 

Academic Editor

PLOS ONE